# Modulation of extrasynaptic GABA$_A$ alpha 5 receptors in the ventral hippocampus normalizes physiological and behavioral deficits in a circuit specific manner

J.J. Donegan[1], A.M. Boley[1], J. Yamaguchi[2], G.M. Toney [2] & D.J. Lodge [1]

Hippocampal hyperactivity is correlated with psychosis in schizophrenia patients and likely attributable to deficits in GABAergic signaling. Here we attempt to reverse this deficit by overexpression of the α5-GABA$_A$ receptor within the ventral hippocampus (vHipp). Indeed, this is sufficient to normalize vHipp activity and downstream alterations in dopamine neuron function in the MAM rodent model. This approach also attenuated behavioral deficits in cognitive flexibility. To understand the specific pathways that mediate these effects, we used chemogenetics to manipulate discrete projections from the vHipp to the nucleus accumbens (NAc) or prefrontal cortex (mPFC). We found that inhibition of the vHipp-NAc, but not the vHipp-mPFC pathway, normalized aberrant dopamine neuron activity. Conversely, inhibition of the vHipp-mPFC improved cognitive function. Taken together, these results demonstrate that restoring GABAergic signaling in the vHipp improves schizophrenia-like deficits and that distinct behavioral alterations are mediated by discrete projections from the vHipp to the NAc and mPFC.

[1] Department of Pharmacology and Center for Biomedical Neuroscience, University of Texas Health Science Center, San Antonio, TX 78229, USA.
[2] Department of Cellular and Integrative Physiology and Center for Biomedical Neuroscience, University of Texas Health Science Center, San Antonio, TX 78229, USA. Correspondence and requests for materials should be addressed to J.J.D. (email: Donegan@uthscsa.edu)

Positive symptoms, such as delusions and hallucinations, are often the most striking features of schizophrenia; however, patients also display characteristic cognitive symptoms, such as working memory deficits and cognitive inflexibility, which can negatively influence social and occupational functioning and diminish quality of life[1–3]. Although the long-standing dopamine hypothesis of schizophrenia suggests that hyperactivity in the mesolimbic dopamine system contributes to disease symptoms[4], antipsychotic medications, which act as antagonists at the dopamine D2 receptor, are only somewhat effective in treating positive symptoms and have little to no impact on cognitive symptoms of schizophrenia[1,5]. Further, schizophrenia patients do not display overt pathology in the mesolimbic dopamine system, leading some to suggest that the pathology of schizophrenia lies in upstream brain regions that regulate dopamine signaling[6].

The hippocampus is one region where functional and anatomical changes have been consistently observed in schizophrenia patients. One of the more reliable observations in schizophrenia patients is increased hippocampal activity at rest[7]. This increase is correlated with positive symptom severity[8] as well as cognitive dysfunction[9], suggesting that exaggerated hippocampal activity may be a key pathogenic factor in schizophrenia. The vHipp hyperactivity observed in schizophrenia patients is thought to result from a deficit in GABAergic inhibition[9]. For example, schizophrenia patients show cell loss restricted to specific GABAergic interneuron subtypes (i.e., parvalbumin and somatostatin) in the hippocampus[10,11]. Previously, we demonstrated that gestational exposure to the mitotoxin methylazoxymethanol (MAM) produces anatomical, physiological, and behavioral deficits that model schizophrenia [for review see ref. [12]], including a loss of hippocampal interneurons[13] and corresponding hippocampal hyperactivity[14], and behavioral correlates of positive[14], negative[15], and cognitive symptoms[15]. We further demonstrated that transplanting interneurons derived from embryonic stem cells, normalizes hippocampal activity and attenuates behavioral correlates of positive and cognitive symptoms in the MAM model[15]. One way in which GABAergic interneurons regulate the function of pyramidal cells in the hippocampus is via the ionotropic GABA$_A$ receptor, a heteropentameric chloride ion channel. The α5 subunit of the GABA$_A$ receptor is unique in its relatively limited distribution within the hippocampus[16]. This subunit has been shown to regulate the timing of pyramidal cell firing, action potential thresholds, and coordinated oscillatory activity[17]. Interestingly, hippocampal specific knock-down of the α5 receptor subunit produces impairments that recapitulate positive symptoms of schizophrenia, including deficits in latent inhibition and pre-pulse inhibition[18,19]. Conversely, systemic α5 agonists normalize dopamine signaling and improve behavioral correlates of positive symptoms in rodent models of schizophrenia[20], suggesting that this subunit of the GABA$_A$ receptor may be a viable therapeutic target. However, it is unclear whether enhancing signaling at the α5 GABA$_A$ subunit would also improve cognitive symptoms, which are poorly treated by currently prescribed antidepressants.

Gene therapy holds the promise of treating diseases by replacing defective genes and has been used to target GABAergic signaling in neurological disorders[21]. In the current experiments, we use viral-mediated gene transfer to restore inhibitory signaling in the vHipp by over-expressing the α5 subunit of the GABA$_A$ receptor in pyramidal cells. We found that α5 overexpression increased tonic GABA currents and normalized aberrant pyramidal cell activity in the vHipp. This approach also normalized aberrant dopamine signaling and cognitive function in a rodent model of schizophrenia, suggesting this may be a promising novel treatment strategy for schizophrenia. Next, we identified the

neural circuit mechanisms underlying these effects. Using chemogenetics, we identified the discrete pathways from vHipp that mediate behavioral and physiological deficits that mirror positive and cognitive symptoms of the disorder. Together, these experiments identify a novel approach for treating schizophrenia and provide insight into the anatomical and neurochemical pathways associated with discrete dimensions of schizophrenia, so that therapeutics can be developed with improved efficacy for treating multiple symptom domains.

## Results

**Effects of α5 overexpression on vHipp neuronal activity.** All data are presented as mean ± SEM. First, to confirm stable transgene expression in the vHipp, we use immunohistochemistry for GFP. As shown in Figure 1a, we observed GFP-labeled cells throughout the pyramidal cell layer of the vHipp. The maximal spread of infection was calculated for a subset of animals to be $2.8 \pm 0.57$ mm from the site of injection. Further, this GFP expression colocalized with CAMKII staining (Fig. 1b), confirming the selectivity of the CAMKII promotor for pyramidal cells in this brain region. The α5 subunit is predominately located extrasynaptically, while the α1 subunit is primarily found in synapses[16,22]. Therefore, we used whole-cell patch-clamp recordings to determine if α1 or α5 overexpression was sufficient to augment tonic or phasic activity. We found that overexpressing the α5 subunit in pyramidal cells of the vHipp increased tonic GABA currents without affecting phasic activity. Specifically, we found that α5 overexpression increased the amplitude of tonic GABA currents (Fig. 1c; one-way ANOVA $F_{(2,18)} = 6.84$, $p < 0.05$; Holm–Sidak GFP vs α5 t = 2.51, $p < 0.05$; GFP $= -19.47 \pm 4.29$ pA, α5 $= -44.16 \pm 10.84$ pA; $n = 6$–9 cells per group) while overexpression of the α1 subunit had no effect on tonic current amplitude (Holm–Sidak GFP vs α1 t = 1.03, $p > 0.05$; α1 $= -9.71 \pm 3.74$ pA). Neither IPSC amplitude nor frequency were affected by either α1 or α5 overexpression in hippocampal slice preparations (Fig. 1c; IPSC amplitude: one-way ANOVA $F_{(2,21)} = 0.30$, $p > 0.05$; Holm–Sidak GFP vs α1 t = 0.07, $p > 0.05$; Holm–Sidak GFP vs α5 t = 0.67, $p > 0.05$; GFP $= -54.07 \pm 4.80$ pA, α1 $= -53.40 \pm 4.78$ pA, α5 $= -60.31 \pm 10.45$ pA; IPSC frequency: one-way ANOVA $F_{(2,21)} = 0.17$, $p > 0.05$; Holm–Sidak GFP vs α1 t = 0.58, $p > 0.05$; Holm–Sidak GFP vs α5 t = 0.16, $p > 0.05$; GFP $= 1.05 \pm 0.28$ Hz, α1 $= 0.83 \pm 0.30$ Hz, α5 $= 0.98 \pm 0.21$ Hz). Representative traces are shown in Fig. 1d and images of recorded cells are shown in (1e) and (1f).

Next, we determined if overexpression of the α1 or α5 subunit of the GABA$_A$ receptor produces functional changes in in vivo vHipp pyramidal cell activity using extracellular electrophysiology. Consistent with the in vitro data, we demonstrate that α5, but not α1, overexpression normalizes the aberrant vHipp activity observed in the MAM model (Fig. 1g; two-way ANOVA: Interaction $F_{(1,326)} = 4.73$, $p < 0.05$; Prenatal Treatment $F_{(1,326)} = 0.10$, $p > 0.05$; Gene Therapy $F_{(1,326)} = 2.40$, $p > 0.05$; $n = 43$–62 cells per group). Specifically, as we have seen previously[15], MAM-treated rats have an increase in pyramidal cell firing rate in the vHipp compared to saline controls (saline/GFP $= 0.68 \pm 0.08$ Hz, MAM/GFP $= 0.89 \pm 0.07$ Hz; Holm–Sidak saline/GFP vs MAM/GFP t = 2.06, $p < 0.05$). In MAM-treated rats that received the virus to over-express the α5 subunit of the GABA$_A$ receptor, this effect was completely abolished (MAM/α 5 $= 0.52 \pm 0.07$ Hz; Holm–Sidak MAM/GFP vs MAM/α5 t = 3.84, $p < 0.05$). In saline-treated rats, α5 overexpression had no effect on pyramidal cell firing rate in the vHipp (saline/α5 $= 0.74 \pm 0.07$ Hz; Holm–Sidak saline/GFP vs saline/α5 t = 0.56, $p > 0.05$). Overexpression of the α1 subunit had no effect in either the saline- or the MAM-treated animals (saline/α1 $= 0.68 \pm 0.07$ Hz,

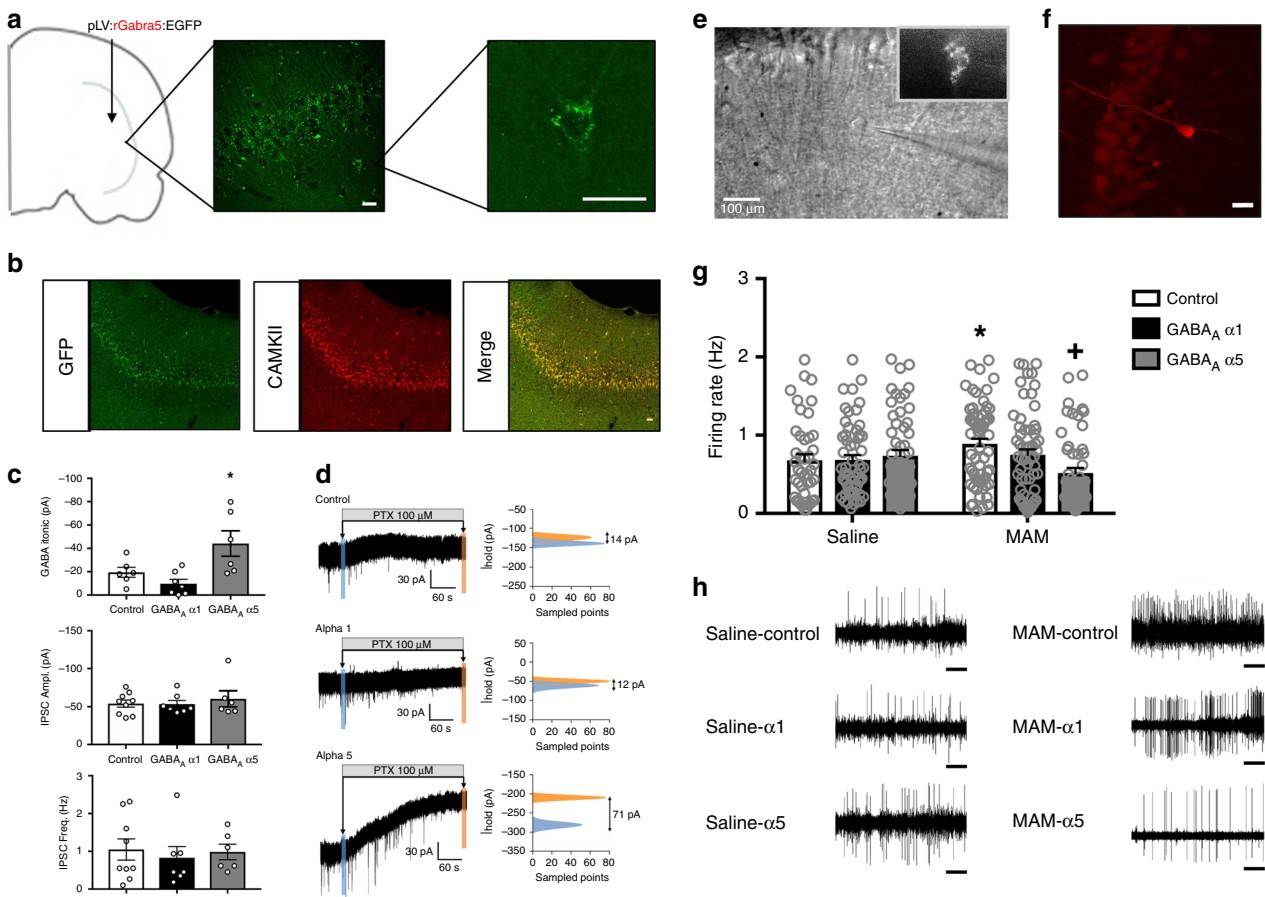

**Fig. 1** α5 overexpression reduces tonic currents and firing rates in vHipp pyramidal cells. The placement of virus injections are indicated on a schematic of a coronal section through the vHipp. Circles indicate injections of the control virus. Squares indicate injections of the GABA_A virus. Immunohistochemistry for GFP was used to confirm transgene expression in the vHipp. Representative images are shown in **a**. To verify that gene expression was confined primarily to CAMKII-positive pyramidal cells, we also performed dual-labelling for GFP and CAMKII. Representative images are shown in **b**. Overexpression of the α5, but not the α1, subunit of the GABA_A receptor increased tonic GABA currents (**c**). Neither α1 nor α5 overexpression affected IPSC frequency or amplitude (**c**). Representative traces are shown in **d**. A representative GFP-positive cell is depicted in **e**. A subset of recorded cells were labeled with neurobiotin and a representative image of a neurobiotin-labeled pyramidal cell is shown in **f**. Scale bars are 20 microns unless otherwise labeled. Asterisk is significantly different than control using One-way ANOVA and Holm–Sidak tests. $n = 6$–9 cells per group. Extracellular electrophysiology was used to measure the firing rates of putative pyramidal cells in the vHipp. In the MAM model of schizophrenia, there is an increase in pyramidal cell firing rate, which is completely abolished by overexpression of the α5, but not α1 subunit of the GABA_A receptor (**g**). Representative traces are shown in **h**. Scale bar is 5 s. Asterisk is significantly different than saline/control; +plus is significantly different than MAM/control using Two-Way ANOVA and Holm–Sidak tests. $n = 43$–62 cells per group. Data are shown as mean ± SEM

MAM/α1 $= 0.75 \pm 0.07$ Hz; Holm–Sidak saline/GFP vs saline/α1 $t = 0.05$, $p > 0.05$; Holm–Sidak MAM/GFP vs MAM/α1 $t = 1.51$, $p > 0.05$). Representative electrophysiology traces are depicted in Fig. 1h. Together, these results suggest that gene therapy can be used to increase α5 expression in pyramidal cells of the vHipp and this overexpression has functional consequences for pyramidal cell activity.

**Effects of α5 overexpression on VTA neuronal activity.** Although it is difficult to model delusions and hallucinations in a rodent, these positive symptoms have been attributed to increases in dopamine neurotransmission[6]. Therefore, we used in vivo extracellular electrophysiology to measure dopamine cell activity as a proxy for positive symptoms and found that overexpression of the α5 subunit of the GABA_A receptor in the vHipp can normalize aberrant dopamine population activity in the VTA (Fig. 2b; two-way ANOVA: Interaction $F_{(1,24)} = 12.43$, $p < 0.05$; Prenatal Treatment $F_{(1,24)} = 19.18$, $p < 0.05$; Gene

Therapy $F_{(1,24)} = 24.25$, $p < 0.05$; $n = 6$–7 rats per group). As we have shown previously[15], MAM-treated animals have an increase in the number of spontaneously active dopamine cells per track compared to saline-treated control animals (saline/GFP $= 1.09 \pm 0.09$ cells/track; MAM/GFP $= 1.77 \pm 0.09$ cells/track; Holm–Sidak saline/GFP vs MAM/GFP $t = 5.49$, $p < 0.05$). This effect is attenuated in MAM-treated animals when the α5 subunit of the GABA_A receptor is over-expressed in pyramidal cells of the vHipp (MAM/α5 $= 1.05 \pm 0.09$ cells/track; Holm–Sidak MAM/GFP vs MAM/α5 $t = 5.87$, $p < 0.05$). Dopamine cell population activity was not affected by α5 overexpression in saline-treated controls (Saline/α5 $= 0.98 \pm 0.08$ cells; Holm–Sidak saline/GFP vs saline/α5 $t = 1.01$, $p > 0.05$). We also analyzed two additional parameters of dopamine cell activity: firing rate and the percentage of action potentials fired in bursts. We found that overexpression of the α5 subunit in the vHipp affected firing rate (Fig. 2c; two-way ANOVA: Interaction $F_{(1,23)} = 2.522$, $p > 0.05$; Prenatal Treatment $F_{(1,23)} = 0.05$, $p > 0.05$; Gene Therapy $F_{(1,23)} = 4.52$, $p < 0.05$). In the

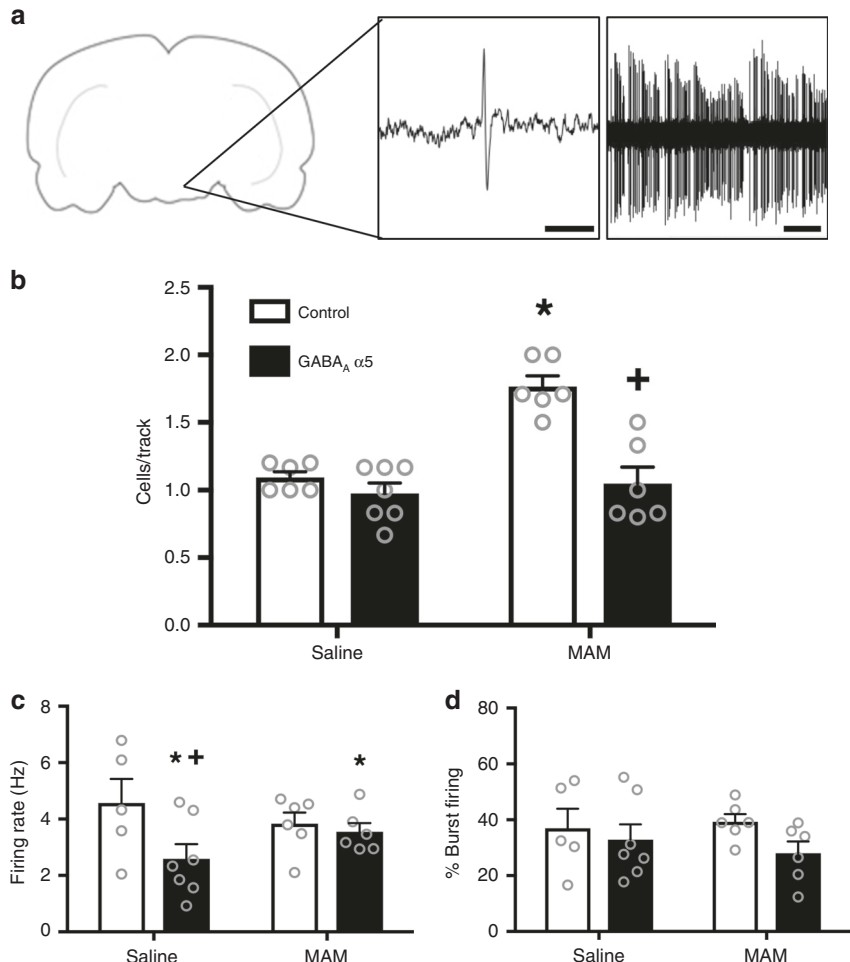

**Fig. 2** α5 overexpression normalizes dopamine cell activity in the MAM model of schizophrenia. Extracellular electrophysiology was used to record dopamine cell activity in the VTA. Representative traces are shown in **a**. In the MAM model of schizophrenia, the increase in the number of spontaneously active dopamine cells is attenuated by overexpression of the α5 subunit in pyramidal cells of the vHipp (**b**). Asterisk is significantly different than saline/control; +plus is significantly different than MAM/control. The average firing rate of dopamine cells was decreased by α5 overexpression in both saline- and MAM-treated animals (**c**). Asterisk is significantly different than saline; +plus is significantly different than saline/control. The percentage of cells firing in a burst pattern was not affected by either MAM treatment or by α5 overexpression (**d**). $n = 6$-7 rats per group. Scale bar is 10 ms. All data were analyzed using Two-Way ANOVA and Holm–Sidak tests. Data are shown as mean ± SEM

saline-treated animals, the α5 overexpression produced a significant decrease in dopamine cell firing rate compared to animals that received the control virus (saline/GFP = 4.57 ± 0.58 Hz; saline/α5 = 2.59 ± 0.49 Hz; Holm–Sidak saline/GFP vs saline/α5 t = 2.608, $p < 0.05$). In the MAM-treated animals, gene therapy had no effect on the firing rate of dopamine neurons (MAM/GFP = 3.84 ± 0.53 Hz; MAM/α5 = 3.55 ± 0.53 Hz; Holm–Sidak MAM/GFP vs MAM/α5 t = 0.38, $p > 0.05$). Overexpression of the α5 subunit of the GABA$_A$ receptor in the vHipp had no effect on the bursting activity of dopamine neurons in the VTA (Fig. 2d; two-way ANOVA: Interaction $F_{(1,23)} = 0.52$, $p > 0.05$; Prenatal Treatment $F_{(1,23)} = 0.06$, $p > 0.05$; Gene Therapy $F_{(1,23)} = 2.35$, $p > 0.05$; saline/GFP = 36.97 ± 5.45% bursting; saline/α5 = 32.89 ± 4.61% bursting; MAM/GFP = 39.33 ± 4.98% bursting; MAM/α5 = 28.05 ± 4.98% bursting). Representative traces are shown in Fig. 2a. Together, these results suggest that gene therapy to increase expression of the α5 subunit of the GABA$_A$ receptor in pyramidal cells of the vHipp normalizes activity in the dopamine system and may attenuate the dopamine-related positive symptoms of schizophrenia.

**α5 overexpression improves cognitive function**. Reversal learning is one form of cognitive flexibility that is disrupted in schizophrenia[23]. Using the attentional set-shifting test, we demonstrated that MAM-treated animals also show a deficit in reversal learning, which was not affected by gene therapy (Fig. 3a; two-way ANOVA: Interaction $F_{(1,22)} = 0.04$, $p > 0.05$; Prenatal Treatment $F_{(1,22)} = 26.28$, $p < 0.05$; Gene Therapy $F_{(1,22)} = 0.62$, $p > 0.05$; $n = 5$–7 rats per group). Specifically, we found that MAM-treated animals show an increase in trials to meet criterion compared to saline-treated controls (saline/GFP = 12.5 ± 1.56 trials; MAM/GFP = 20.40 ± 1.71 trials; Holm–Sidak saline/GFP vs MAM/GFP t = 3.42, $p < 0.05$). The MAM-induced deficit on reversal learning was not attenuated by overexpression of the α5 subunit of the GABA$_A$ receptor (saline/α5 = 13.43 ± 1.44 trials; MAM/α5 = 22.0 ± 1.71 trials; Holm–Sidak saline/α5 vs MAM/α5 t = 3.84, $p < 0.05$). Further, α5 overexpression had no effect on reversal learning in the saline-treated group (Holm–Sidak saline/GFP vs saline/α5 t = 0.44, $p > 0.05$).

In addition to reversal learning deficits, schizophrenia patients also show deficits in extradimensional set-shifting[23], a higher order form of cognitive flexibility that is critically dependent on

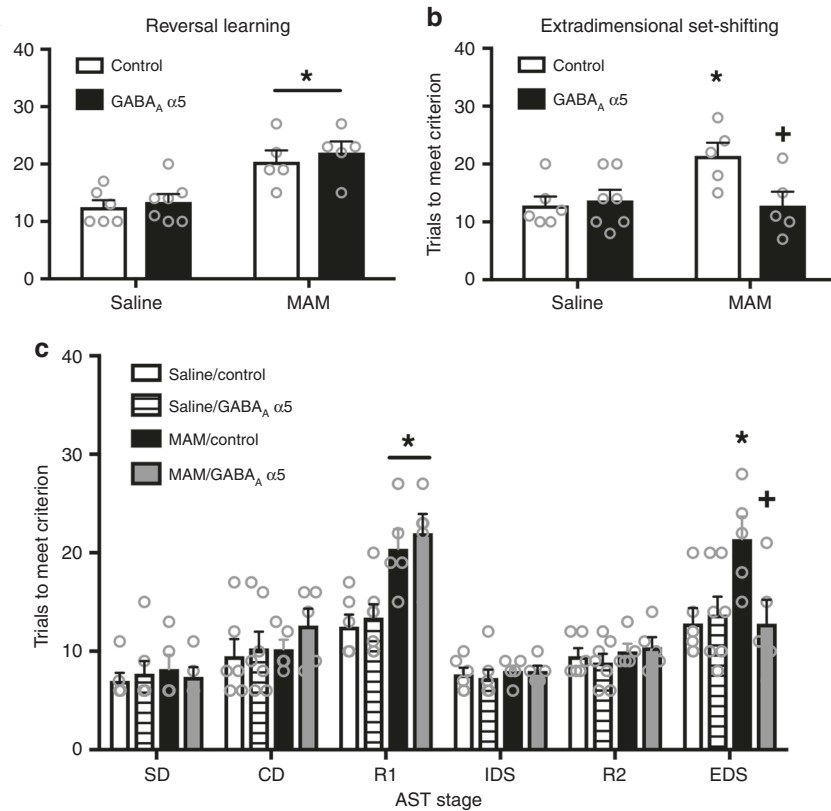

**Fig. 3** α5 overexpression improves some forms of cognitive flexibility in the MAM model. The attentional set-shifting test (AST) was used to measure reversal learning and extradimensional set-shifting (ED), two forms of cognitive flexibility. The MAM-induced deficit in reversal learning was not affected by over-expressing the α5 subunit in the vHipp (**a**). Asterisk is a significant main effect compared to saline. The deficit in extradimensional set-shifting caused by MAM treatment is completely abolished by gene therapy to over-express the α5 subunit of the GABA$_A$ receptor (**b**). Asterisk is significantly different than saline/control; +plus is significantly different than MAM/control. All stages of the AST are shown in **c**. SD is simple discrimination. CD is compound discrimination. R1 is the first reversal. IDS is intradimensional set-shift. R2 is the second reversal. EDS is the extradimensional set-shift. $n = 5$–7 rats per group. All data were analyzed using Two-Way ANOVA and Holm–Sidak tests. Data are shown as mean ± SEM

the medial prefrontal cortex (mPFC)[24]. Unlike reversal learning, we found that gene therapy to over-express the α5 subunit of the GABA$_A$ receptor improves schizophrenia-like deficits in extradimensional set-shifting as measured by the attentional set-shifting test (Fig. 3b; two-way ANOVA: Interaction $F_{(1,21)} = 6.02$, $p < 0.05$; Prenatal Treatment $F_{(1,21)} = 2.92$, $p > 0.05$; Gene Therapy $F_{(1,21)} = 2.97$, $p > 0.05$; $n = 5$–7 rats per group). As we have shown previously[15], MAM-treated animals have a deficit in extradimensional set-shifting as evidenced by an increase in trials to meet criterion compared to controls (saline/GFP = 12.83 ± 1.96 trials; MAM/GFP = 21.40 ± 2.15 trials; Holm–Sidak saline/GFP vs MAM/GFP t = 2.94, $p < 0.05$). This deficit was completely abolished in the animals that received gene therapy to over-express the α5 subunit in pyramidal cells of the vHipp (saline/α5 = 14.33 ± 1.96 trials; MAM/ α5 = 12.8 ± 2.15 trials; Holm–Sidak saline/α5 vs MAM/α5 t = 0.53, $p > 0.05$). The α5 overexpression had no effect in the saline-treated controls (Holm–Sidak saline/GFP vs saline/α5 t = 0.54, $p > 0.05$). No other stages of the AST were affected by MAM treatment or α5 overexpression (Fig. 3c), suggesting that there was not a deficit in basic learning or memory processes. Together, these results suggest that gene therapy to increase α5 expression in the vHipp may be an effective treatment strategy to alleviate both positive and cognitive deficits associated with schizophrenia.

**The vHipp-NAc pathway mediates aberrant dopamine activity.** We have previously demonstrated that the increase in dopamine

population activity is directly attributable to a pathologically enhanced drive from the vHipp[14]. However, the vHipp does not project directly to the VTA; therefore, in the current experiments, we demonstrate that reducing activity in the vHipp-NAc pathway of MAM-treated animals abolishes the MAM-induced increase in dopamine population activity (Fig. 4f; two-way ANOVA: Interaction $F_{1,19} = 15.98$, $p < 0.05$; Prenatal treatment $F_{1,19} = 12.04$, $p < 0.05$; DREADD condition $F_{1,19} = 6.607$, $p < 0.05$; $n = 4$–5 rats per group). In line with our previous findings, MAM-treated animals show a significant increase in the number of spontaneously active dopamine cells per track (saline/GFP = 0.90 ± 0.14 cells per track, MAM/GFP = 1.95 ± 0.14 cells per track; Holm–Sidak saline/GFP vs MAM/GFP t = 5.28, $p < 0.05$). Chemogenetic inhibition of the vHipp-NAc pathway attenuated the increase in dopamine population activity in MAM-treated animals but had no effect in controls (saline/Gi = 1.10 ± 0.14 cells per track, MAM/Gi = 1.03 ± 0.14 cells per track; Holm–Sidak MAM/GFP vs MAM/Gi: t = 4.64, $p < 0.05$; Holm–Sidak saline/GFP vs saline/Gi: t = 1.01, $p > 0.05$). The firing rate (Fig. 4g; two-way ANOVA: Interaction $F_{(1,19)} = 1.161$, $p > 0.05$; Prenatal treatment $F_{(1,19)} = 0.52$, $p > 0.05$; DREADD condition $F_{(1,19)} = 0.59$, $p > 0.05$; saline/GFP = 2.6 ± 0.61 Hz, saline/Gi = 3.73 ± 0.61 Hz; MAM/GFP = 3.70 ± 0.61 Hz, MAM/Gi = 3.51 ± 0.61 Hz) and burst pattern (Fig. 4h; two-way ANOVA: Interaction $F_{(1,19)} = 1.74$, $p > 0.05$; Prenatal treatment $F_{(1,19)} = 0.10$, $p > 0.05$; DREADD condition $F_{(1,19)} = 0.65$, $p > 0.05$; saline/GFP = 65.83 ± 11.12%, saline/Gi = 42.22 ± 11.12%; MAM/GFP = 47.58 ±

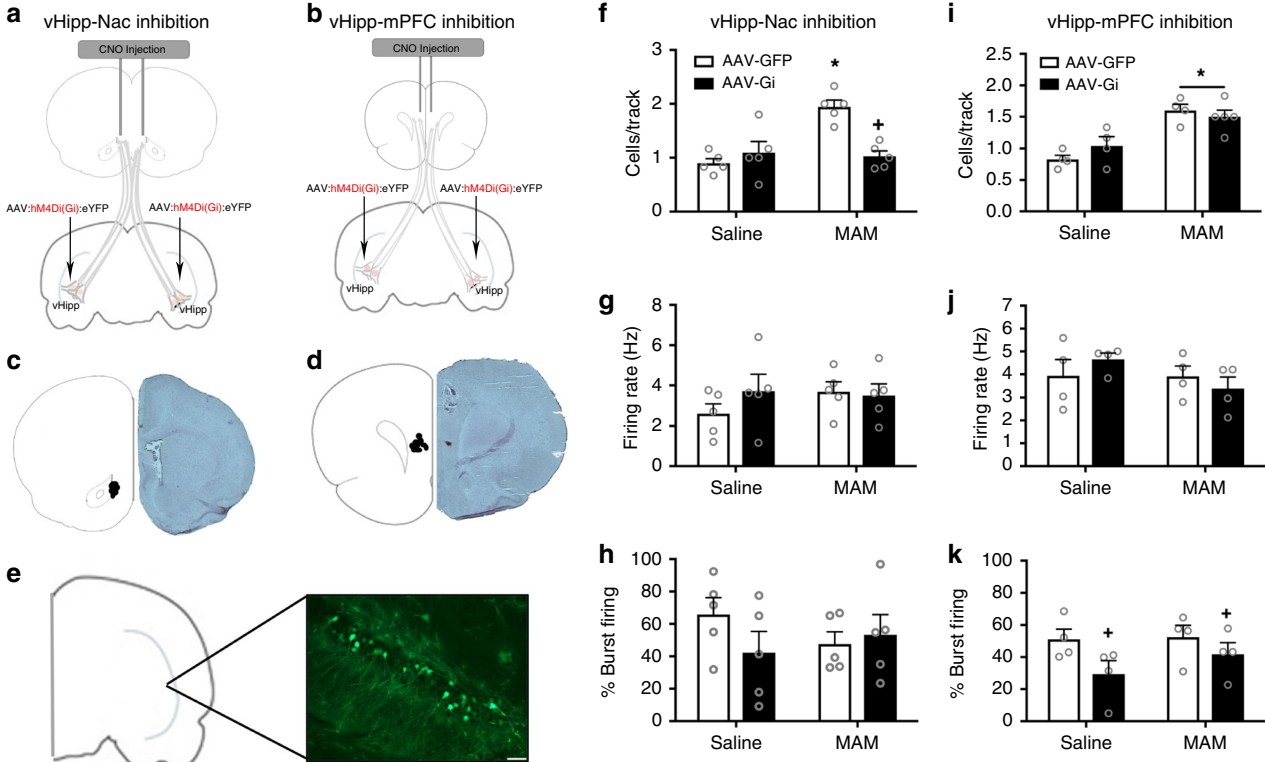

**Fig. 4** Inactivation of the vHipp-NAc normalizes dopamine population activity in MAM-treated rats. The schematic depicting the strategy used for pathway inhibition is shown in **a, b**. An adeno-associated virus that expressed the Gi DREADD was injected into the vHipp. At the time of testing, CNO was injected directly into the NAc (**a**) or mPFC (**b**). Cartoon showing the location of the CNO injection and micrograph of representative section are shown in **c, d**. DREADD expression in pyramidal cells of the vHipp was confirmed using immunohistochemistry and a representative image is shown in **e**[70]. n = 3 rats per group. Scale bar is 50 microns. MAM-treated animals show an increase in the number of spontaneously active dopamine cells per track in the VTA compared to saline controls. Chemogenetic inactivation of vHipp afferents to the NAc completely abolishes the MAM-induced increase in dopamine population activity (**f**). Neither firing rate (**g**) nor the percentage of action potentials fired in bursts (**h**) were affected by MAM treatment or pathway inhibition. n = 5 rats per group. Asterisk is significantly different from saline/GFP; +plus is significantly different than MAM/GFP. The MAM-induced increase in dopamine cell population activity is not affected by inhibition of the vHipp to mPFC pathway (**i**). Neither MAM treatment nor pathway inhibition affect firing rate (**j**). The percentage of action potentials fired in a burst pattern was decreased by vHipp-mPFC pathway inhibition in both the saline- and MAM-treated animals (**k**). n = 4–5 rats per group. Asterisk is significantly different from saline; +plus is significantly different than GFP. All data were analyzed using Two-Way ANOVA and Holm–Sidak tests. Data are shown as mean ± SEM

11.12%, MAM/Gi = 53.29 ± 11.12%) of VTA dopamine cells were not affected by either MAM treatment or pathway inhibition.

Conversely, inhibition of the vHipp-mPFC pathway has no effect on dopamine cell firing in the VTA (Fig. 4i; two-way ANOVA: Interaction $F_{1,16} = 2.17$, $p > 0.05$; Prenatal treatment $F_{1,16} = 32.76$, $p < 0.05$; DREADD condition $F_{1,16} = 0.29$, $p > 0.05$; n = 4–5 per group). Although MAM-treated animals showed a significant increase in the number of spontaneously active dopamine cells per track (saline/GFP = 0.83 ± 0.11 cells per track, MAM/GFP = 1.60 ± 0.11 cells per track; Holm–Sidak saline/GFP vs MAM/GFP t = 4.96, $p < 0.05$), inhibition of the vHipp-mPFC pathway did not normalize dopamine population activity (saline/ Gi = 1.04 ± 0.11 cells per track, MAM/Gi = 1.50 ± 0.10 cells per track; Holm–Sidak MAM/GFP vs MAM/Gi t = 0.68, $p > 0.05$). Firing rate was not affected by either prenatal treatment or pathway inhibition (Fig. 4J; two-way ANOVA: Interaction $F_{(1,15)} = 1.508$, $p > 0.05$; Prenatal treatment $F_{(1,15)} = 1.62$, $p > 0.05$; DREADD condition F = 0.03, $p > 0.05$; saline/GFP = 3.93 ± 0.51 Hz, MAM/GFP = 3.91 ± 0.51 Hz, Saline/Gi = 4.66 ± 0.51 Hz, MAM/Gi = 3.37 ± 0.51 Hz). There was a main effect of the Gi DREADD on the percentage of action potentials fired in bursts (Fig. 4k; two-way ANOVA: Interaction $F_{(1,15)} = 0.55$, $p > 0.05$; Prenatal treatment $F_{(1,15)} = 0.87$, $p > 0.05$; DREADD condition

F = 4.79, $p < 0.05$; saline/GFP = 51.01 ± 7.38%, MAM/GFP = 52.41 ± 7.38%, Saline/Gi = 29.37 ± 7.38%, MAM/Gi = 41.74 ± 7.38%). Cannula placements are shown in Fig. 4c, d and a representative image of DREADD expression is shown in Fig. 4e.

Together, these results suggest that hyperactivity in the vHipp-NAc pathway, but not the vHipp-mPFC pathway, underlies the increase in dopamine cell population activity observed in the MAM model of schizophrenia.

**The vHipp-NAc pathway mediates deficits in reversal learning.** To identify the neural pathways involved in unique forms of cognitive flexibility, we used the attentional set-shifting test. We found that hyperactivity in the vHipp-NAc pathway is responsible for reversal learning deficits in the MAM model of schizophrenia (Fig. 5a; two-way ANOVA: Interaction $F_{(1,26)} = 1.68$, $p > 0.05$; Prenatal treatment $F_{(1,26)} = 10.11$, $p < 0.05$; DREADD Condition $F_{(1,26)} = 8.38$, $p < 0.05$; n = 6–7 per group). Specifically, we found that MAM treatment produced a deficit in reversal learning, as evidenced by an increase in trials to meet criterion (saline/GFP = 14.67 ± 1.77 trials to meet criterion, MAM/GFP = 22.14 ± 1.64 trials; Holm–Sidak saline/GFP vs MAM/GFP t = 3.10, $p < 0.05$). The reversal learning deficit was

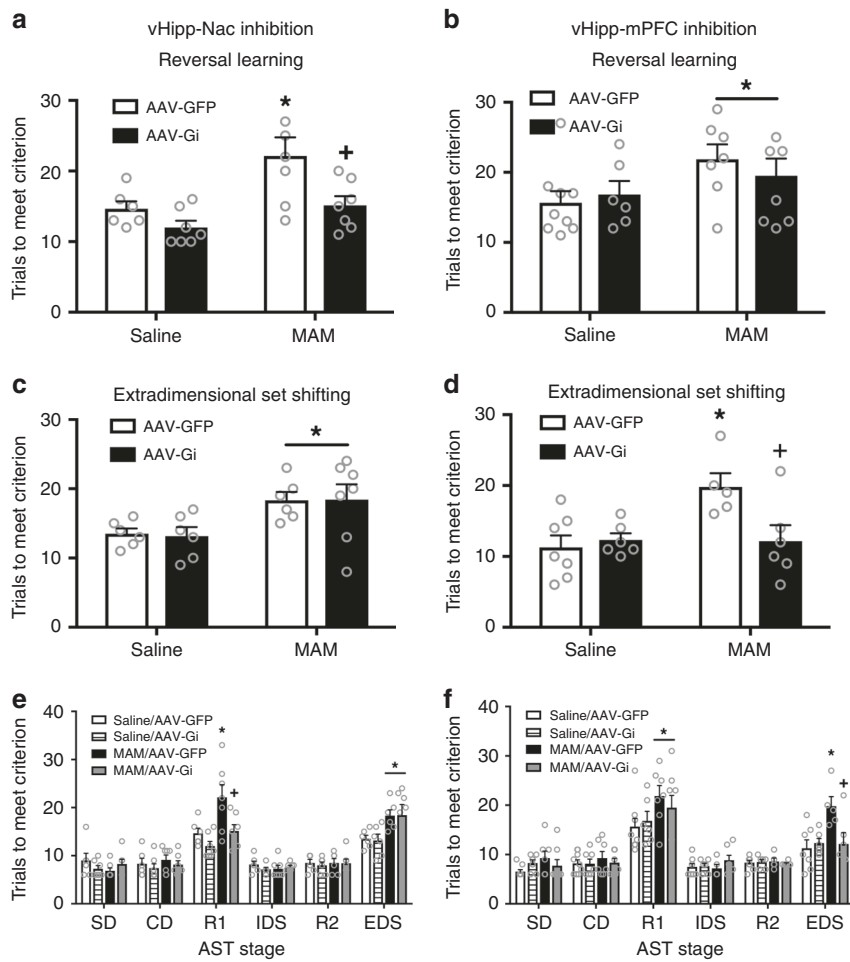

**Fig. 5** vHipp-NAc and vHipp-mPFC pathways differentially affect distinct types of cognitive flexibility. MAM treatment causes a deficit in both reversal learning and in extradimensional set-shifting. Inhibition of the vHipp-NAc pathway completely abolishes the MAM-induced deficit in reversal learning (**a**). Asterisk is significantly different than saline/GFP. Plus is significantly different than MAM/GFP. However, vHipp-NAc inhibition had no effect on extradimensional set-shifting (**c**). Asterisk is significant main effect of MAM. $n = 6$–7 rats per group. Conversely, inhibition of the vHipp-mPFC pathway had no effect on reversal learning performance (**b**) Asterisk is significant main effect of MAM. $n = 6$–9 rats per group. However, vHipp-mPFC inhibition did attenuate the MAM-induced deficit in extradimensional set-shifting (**d**). Aserisk is significantly different than saline/GFP; plus is significantly different than MAM/GFP. $n = 6$–7 rats per group. All stages of the AST are shown in **e**, (**f**). All data were analyzed using Two-Way ANOVA and Holm–Sidak tests. Data are shown as mean ± SEM

attenuated by inhibition of the vHipp-NAc pathway (MAM/Gi = 15.14 ± 1.64; Holm–Sidak MAM/GFP vs MAM/Gi t = 3.03, $p < 0.05$; Holm–Sidak Saline/Gi vs MAM/Gi t = 1.36, $p > 0.05$). Inhibition of the vHipp-NAc pathway had no effect on reversal learning performance in the saline-treated rats (saline/Gi = 12.00 ± 1.64; Holm–Sidak saline/GFP vs saline/Gi t = 1.11, $p > 0.05$).

Conversely, we found that reversal learning was not affected by inhibition of the vHipp-mPFC pathway (Fig. 5b; two-way ANOVA: Interaction $F_{(1,29)} = 0.71$, $p > 0.05$; Prenatal treatment $F_{(1,29)} = 4.46$, $p < 0.05$; DREADD condition $F_{(1,29)} = 0.0806$, $p > 0.05$; $n = 6$–7 per group). MAM-treated rats required significantly higher trials to reach criterion than saline-treated controls (saline/GFP = 15.67 ± 1.89 trials, MAM/GFP = 21.86 ± 2.145 trials; Holm–Sidak saline/GFP vs MAM/GFP t = 0.04, $p < 0.05$). The MAM-induced deficit was not altered by inhibition of the vHipp-mPFC pathway (saline/Gi = 16.83 ± 2.32 trials, MAM/Gi = 19.50 ± 2.01 trials; Holm–Sidak saline/Gi vs MAM/Gi t = 0.87, $p > 0.05$). Together, these results suggest that hyperactivity in the vHipp-NAc pathway, but not the vHipp-mPFC pathway contributes to the reversal learning deficits observed in the MAM model of schizophrenia.

**The vHipp-mPFC pathway mediates set-shifting deficits.** Unlike reversal learning, we found that extradimensional set-shifting deficits are not affected by inhibition of the vHipp-NAc pathway (Fig. 5c; two-way ANOVA: Interaction $F_{(1,24)} = 0.02$, $p > 0.05$; Prenatal treatment $F_{(1,24)} = 10.49$, $p < 0.05$; DREADD Condition $F_{(1,24)} = 0.01$, $p > 0.05$; $n = 6$–7 per group). As seen previously, we found that MAM-treated animals have a deficit in extradimensional set-shifting as evidenced by an increase in trials to meet criterion (saline/GFP = 13.50 ± 1.59 trials to meet criterion, MAM/GFP = 18.33 ± 1.59 trials; Holm–Sidak saline/GFP vs MAM/GFP t = 2.16, $p < 0.05$). Inactivation of the vHipp-NAc pathway had no effect on extradimensional set-shifting in either the saline- or MAM-treated groups (saline/Gi = 13.17 ± 1.59 trials, MAM/Gi = 18.43 ± 1.47 trials; Holm–sidak saline/GFP vs saline/Gi t = 0.15, $p > 0.05$; Holm–Sidak MAM/GFP vs MAM/Gi t = 0.04, $p > 0.05$).

Conversely, hyperactivity in the vHipp-mPFC pathway is responsible for MAM-induced deficits in extradimensional set-shifting (Fig. 5d; two-way ANOVA: Interaction $F_{(1,23)} = 5.99$, $p < 0.05$; Prenatal treatment $F_{(1,23)} = 5.54$, $p < 0.05$; DREADD condition $F_{(1,23)} = 3.45$, $p > 0.05$; $n = 6$–7 per group). As expected, MAM-treated animals showed a deficit in extradimensional

set-shifting (saline/GFP = 11.286 ± 1.63, MAM/GFP = 19.8±1.93; Holm–Sidak saline/GFP vs MAM/GFP t = 3.37, $p < 0.05$). Although inhibition of the vHipp-mPFC pathway had no effect in saline-treated animals, it completely abolished the MAM-induced deficit in extradimensional set-shifting (saline/Gi = 12.33 ± 1.76, MAM/Gi = 12.17 ± 1.76; Holm–Sidak saline/GFP vs saline/Gi t = 0.44, $p > 0.05$; Holm–Sidak MAM/GFP vs MAM/Gi t = 2.92, $p < 0.05$). Performance on all stages of the AST are shown in Fig. 5e, f. Together, these results suggest that hyperactivity in the vHipp-mPFC pathway, but not the vHipp-NAc pathway, contributes to the schizophrenia-like deficits in extradimensional set-shifting observed in MAM-treated animals.

## Discussion

We, and others, have shown previously that hyperactivity in the ventral hippocampus (vHipp) is responsible for schizophrenia-like deficits in a variety of rodent models[14,25]. Furthermore, normalizing hippocampal activity by transplanting GABAergic interneurons can attenuate behavioral deficits that model positive and cognitive symptoms of the disorder[15]. In the current study, we utilized a virally-mediated genetic approach to demonstrate that targeting tonic GABA signaling in pyramidal cells of the vHipp can normalize aberrant hippocampal activity and attenuate schizophrenia-like deficits. Specifically, we demonstrate that overexpression of the α5 subunit of the GABA$_A$ receptor in pyramidal cells of the vHipp attenuates dopamine cell activity, which is related to positive symptoms of schizophrenia, and reverses behavioral deficits in cognitive flexibility. Further, we identified a potential mechanism by which vHipp pyramidal cell hyperactivity may affect distinct symptom domains associated with schizophrenia and other psychiatric diseases. Specifically, hyperactivity in projections from the vHipp to the NAc contribute to the aberrant increases in dopamine cell activity and is responsible for deficits in dopamine-dependent behaviors. Conversely, hyperactivity in projections to the mPFC have no effect on dopamine cell activity or related behaviors, but are responsible for some forms of cognitive inflexibility. Together, our results suggest that using gene therapy to restore GABAergic signaling in pyramidal cells of the vHipp may reduce activity in projections from the vHipp and may be a viable treatment strategy for targeting multiple symptom domains of schizophrenia.

Gene therapy, the method of using a viral vector to insert a gene directly into a cell, has shown great promise in recent years to not just treat symptoms but to actually cure certain diseases. For example, in Parkinson's Disease (PD), adeno-associated virus was used to over-express the GABA synthesizing enzyme, glutamic acid decarboxylase, in the subthalamic nucleus. The unilateral infusion reduced metabolism in the thalamus and ipsilateral motor and premotor cortices, which was correlated with improved clinical disability ratings. These effects were still apparent at 1 year and were not associated with severe side effects[26]. In the current experiments, we used a lentiviral vector to over-express the α5 subunit of the GABA$_A$ receptor. Lentivirus has also been used in PD patients to express tyrosine hydroxylase, GTP cyclohydrolase, and aromatic acid decarboxylase, three enzymes needed for dopamine synthesis, in striatal neurons. This treatment resulted in an improved motor score for up to 1 year[27]. Further, in the current experiments, we demonstrate that we can improve multiple symptom domains by targeting a single brain region, the vHipp, reducing the possibility of negative side effects. However, further investigation of the long-term safety will be required before this treatment strategy can be moved to the clinic.

In the current experiments, we used viral-mediated gene transfer to increase expression of the α5 subunit of the GABA$_A$ receptor. GABA$_A$ receptors are heteropentameric chloride ion

channels with 19 known subunits (α1–6, β1–3, γ1–3, δ, ε, π, θ, ρl–3). The α5 subunit is unique as its expression is primarily limited to the hippocampus[16], making it an ideal target for specifically reducing the hippocampal hyperactivity associated with schizophrenia. Further, the α5 subunit has been identified primarily extrasynaptically in dendritic fields[16], where it is thought to mediate tonic inhibitory currents[28]. Indeed, in the current experiments, we demonstrate that over-expressing the α5 subunit increases tonic inhibitory currents in pyramidal cells of the vHipp. Further, we show that α5 overexpression also decreases the firing rate of hippocampal pyramidal cells, which is not surprising as tonic inhibition is thought to coordinate spike timing of pyramidal neurons and balance excitation[17]. By over-expressing the α5 subunit of the GABA$_A$ receptor, we normalized pyramidal cell activity in the vHipp.

In addition, we demonstrated that α5 overexpression normalizes VTA dopamine cell population activity in the MAM model. Hyperactivity in the dopamine system is thought to underlie the positive symptoms of schizophrenia. For example, antipsychotic medications that block dopamine D2 receptors effectively reduce positive symptoms of the disorder[29]. Further, elevated dopamine levels have been observed in the striatum of schizophrenia patients, and these levels have been correlated to the severity of positive symptoms[30]. Drugs that increase dopamine signaling can induce psychotic episodes in schizophrenia patients[31]. The current results are in line with previous work demonstrating that administration of the α5 partial agonist, SH-053–2'F-R-CH3, normalizes the number of spontaneously active dopamine neurons and amphetamine-induced locomotor activity in MAM rats[20]. Further, others have shown that α5 knock-out mice exhibit deficits in sensorimotor gating and latent inhibition[18,19]. Together, this data suggest that increasing α5 expression in the vHipp may act through the dopamine system to alleviate positive symptoms of schizophrenia. It is important to note that in animal models of schizophrenia, positive symptoms seem to result from increases in dopamine signaling specifically in ventral regions of the striatum, including the NAc[32]. However, in human studies, imaging techniques, such as positron emission tomography, have demonstrated that the largest changes in dopamine signaling are actually observed in associative regions of the striatum[33]. The reason for this discrepancy is a current area of research that remains to be elucidated.

In addition to dopamine cell activity, we also tested the ability of α5 subunit overexpression to improve schizophrenia-like deficits in cognitive function. We were surprised to find that α5 overexpression had no effect on reversal learning, one form of cognitive flexibility that is disrupted in schizophrenia. Reversal learning has been associated with dopamine signaling. For example, reversal learning performance is correlated with striatal dopamine synthesis capacity[34] and striatal D2 receptor availability[35]. Dopamine depletion in the caudate nucleus impairs reversal learning[36]. Further, both pharmacological agonism[37] and antagonism[38] of D2/D3 receptors can disrupt reversal learning, suggesting that this type of cognitive flexibility requires tightly controlled dopamine signaling. Because gene therapy normalized dopamine cell activity, we expected to see an improvement in reversal learning performance. Conversely, we did find that our gene therapy strategy attenuated the schizophrenia-like deficit in attentional set-shifting, a higher order form of cognitive flexibility mediated primarily by the medial prefrontal cortex (mPFC)[24]. These results suggest that gene therapy to over-express the α5 subunit of the GABA$_A$ receptor in pyramidal cells of the vHipp may be a viable treatment strategy to target some cognitive symptoms of schizophrenia, which have a profound impact on daily function[1–3] but are poorly treated by currently available antipsychotic medications[3,5].

In the current experiments, we chose to focus on the hippo-campus as both in schizophrenia patients and in animal models of schizophrenia, the hippocampus has been shown to be a key site of pathology. Decreases in hippocampal volume have been observed in schizophrenia patients at the time of first episode and seem to progress throughout the course of the illness[39]. These decreases in volume are primarily limited to the anterior hippo-campus[40], the sub-region of the hippocampus that corresponds to the ventral hippocampus of rodents[41,42]. Further, unmedicated schizophrenia patients show increased hippocampal activity at rest[7], an effect that is attenuated by antipsychotic treatment[43]. Work in multiple animal models of schizophrenia has also demonstrated hyperactivity in the ventral hippocampus. For example, vHipp hyperactivity has been observed in develop-mental[14], genetic[44], and pharmacological models [25]of schizo-phrenia. Further, we and others have shown that reducing hippocampal activity by transplanting inhibitory interneurons can normalize positive and cognitive symptoms of the disorder[15,44]. In the current studies, we expanded on these findings to identify the neural circuits from the vHipp that influence specific aspects of schizophrenia-like behavior. The concept that unique symptom clusters can arise from disruptions in subcortical structures, which then lead to abnormal regulation of cortical activity and dopamine system function, has been previously proposed by O'Donnell and Grace[45]. In order to test this hypothesis by manipulating discrete pathways from the vHipp we used chemogenetics, a method in which small molecule chemical actuators specifically interact with engineered proteins to affect the activity of a cell, such as G-protein-coupled receptors (GPCRs)[46]. Specifically, we used the Designer Receptor Exclu-sively Activated by Designer Drugs (DREADD), hM4Di, a modified M4 muscarinic acetylcholine receptor that couples to the inhibitory Gi pathway when activated by clozapine-N-oxide, a highly selective, but pharmacologically inert, exogenous ligand[47]. The hM4Di DREADD has been shown to silence neurons by hyperpolarizing the cell via activation of G-protein inwardly rectifying potassium channels (GIRKs)[47,48] and inhibiting pre-synaptic neurotransmitter release[49]. In the current experiments, we used an adeno-associated viral vector to over-express the inhibitory DREADD in cells of the vHipp, then injected CNO directly into the NAc or mPFC to specifically inhibit vHipp neurons that project to these regions. Although CNO is often administered systemically to inhibit activity in discrete brain regions, this pathway-targeting strategy has been used success-fully by others[50].

One way in which the vHipp may influence schizophrenia-like behavior is through its control of the dopamine system[45]. In rodents, the vHipp can regulate dopamine signaling via a poly-synaptic pathway from the NAc to the ventral pallidum (VP), and ultimately to dopamine cells in the ventral tegmental area (VTA)[51]. Specifically, the vHipp regulates the dopamine neuron population activity, or the number of spontaneously active dopamine neurons, which has been correlated with dopamine efflux in the NAc[32]. Importantly, it has been shown that only spontaneously active dopamine neurons can respond to stimuli by firing in a bursting pattern[52]. In this way, the vHipp is thought to regulate the 'gain' of the dopamine system[6]. In the current studies, we demonstrate that hyperactivity in the vHipp-NAc pathway of MAM animals increases dopamine population activity in the VTA by showing a reduction in the number of sponta-neously active dopamine cells per track after inhibition of the vHipp-NAc pathway.

Interestingly, we also found that inhibition of the vHipp-NAc pathway attenuates MAM-induced deficits in reversal learning. This is in contrast to the alpha 5 overexpression experiment, which also normalized dopamine cell activity in the VTA but did not improve reversal learning in MAM-treated rats. Therefore, it is unlikely that inhibition of the vHipp-NAc pathway in MAM-treated animals improved reversal learning by normalizing dopamine activity in the striatum. However, other neuro-transmitter systems and brain circuits have been implicated in reversal learning, including serotonin signaling in the orbital frontal cortex (OFC)[23]. Furthermore, impacting the OFC-NAc pathway could occur either by normalization of vHipp-NAc activity or normalization of VTA-NAc activity. Further experi-ments are required to conclusively determine which specific pathways and neurotransmitters are associated with the effect of vHipp-NAc on reversal learning.

In addition to its projection to the NAc, the vHipp also projects directly to the prefrontal cortex[53], a brain region that has been implicated in the cognitive symptoms of schizophrenia[54]. Schi-zophrenia patients show reduced prefrontal cortex activation during working memory tasks, with activation levels correlated to performance[55]. Even schizophrenia patients that perform nor-mally on working memory tasks show higher levels of PFC activation compared to controls, suggesting a deficit in effi-ciency[56]. Further, anatomical changes, such as reduced dendritic spine density have been observed the PFC of schizophrenia patients[57]. In line with these findings, we demonstrated that reducing hyperactivity in the vHipp-mPFC pathway improved attentional set-shifting performance in MAM-treated rats. This is not surprising as the mPFC has been implicated in this form of cognitive flexibility[24].

In the current experiments, we did not observe an effect of vHipp-mPFC pathway inhibition on dopamine population activity. However, it should be noted that we have previously demonstrated that synchronous cortical burst firing of the mPFC can increase dopamine population activity in the VTA[58]. These studies also demonstrated that tonic activation of the mPFC produced subtle changes in VTA dopamine neurons suggesting that the effects of mPFC inhibition on VTA dopamine neuron activity would only be observed during periods of cortical burst firing.

In the current experiments, we focused on pathways from the vHipp to the NAc and mPFC. However, schizophrenia is a het-erogeneous disorder and the vHipp is not the only site of pathology. For example, structural and functional changes have also been observed in the thalamus of schizophrenia patients[59] and we have recently demonstrated that the paraventricular nucleus of the thalamus can also regulate dopamine signaling via the NAc[60]. Further, the vHipp sends and receives many projec-tions to and from other brain regions beyond the NAc and mPFC. For example, reciprocal connections exist between the basolateral amygdala (BLA) and vHipp[61] and optogenetic inhi-bition of this BLA-vHipp pathway has been shown to decrease anxiety-like behavior[62] and increase social interaction time[63], which has been used to model negative symptoms of the disorder. Conversely, activation of the pathway increases anxiety-like behaviors[62] and decreases social interaction time[63]. The BLA has been implicated in schizophrenia[64], therefore, it is likely that changes in this neural circuit may also contribute to the pathol-ogy of schizophrenia. However, schizophrenia is a complex dis-order and the individual pathways examined do not exist in isolation. Further, each patient experiences a unique set of symptoms. Therefore, we and other believe that symptom clusters in schizophrenia result from disruptions in interconnected neural systems involving the pathways examined in the current experiments[45].

In conclusion, we have demonstrated that normalizing hip-pocampal activity using a gene therapy strategy to over-express the α5 subunit of the $GABA_A$ receptor in pyramidal cells can attenuate schizophrenia-like deficits in a rodent model. Further,

we showed that discrete projections from the vHipp differentially mediate symptoms of the disorder. We believe that by better understanding the neuronal pathways associated with discrete dimensions of antipsychotic efficacy, novel therapeutics can be developed with improved efficacy at treating multiple symptom domains associated with schizophrenia.

## Methods

**Animals**. All experiments were performed in accordance with the guidelines outlined in the USPH Guide for the Care and Use of Laboratory Animals and were approved by the Institutional Animal Care and Use Committee of the University of Texas Health Science Center at San Antonio. Rats were maintained on a 12 h/12 h light/dark cycle, with food and water available *ad libitum* unless specified below.

**MAM administration**. To model circuit level alterations and behavioral deficits associated with schizophrenia, timed pregnant female Sprague–Dawley rats were obtained from Harlan on gestational day 16. Methylazoxymethanol (22 mg/kg i.p.) or saline was administered on gestational day 17, a dose and time point that induces schizophrenia-like changes in behavior and neuronal activity[12]. Male pups were weaned on postnatal day 21 and housed in groups of 3 until they were >12 weeks old, at which point rats were singly housed and used for behavioral or electrophysiological experiments. All experiments included pups from multiple litters.

**Stereotaxic surgeries**. To over-express the α1 or α5 subunit of the GABA$_A$ receptor under the control of the CAMKII promoter, lentiviral vectors were used. On postnatal days 40–45, animals were anesthetized using Fluriso (2–5% Isoflurane, USP with oxygen flow at 1 L/min) and placed in a stereotaxic apparatus. Bilateral cannula aimed at the vHipp (A/P + 4.8, M/L ± 4.8, D/V −6.0 mm from bregma as determined using[70]) were used to inject the α1 virus (pLV-CaMKII-rGabra1-IRES-EGFP; 3.19 × 10$^9$ TU/ml), α5 virus (pLV-CaMKII-rGabra5-IRES-EGFP; 2.86 × 10$^9$ TU/ml) or the control virus (pLV-EGFP:T2A:puro-EF1A > mcherry; 1.26 × 10$^9$ TU/ml) into each hemisphere (1.0 ul; VectorBuilder). Animals were allowed to recover for >6 weeks before behavioral and electrophysiological experiments to allow maximal gene expression.

To over-express the designer receptor hM4D (Gi), a modified version of the human Gi-coupled muscarinic receptor 4 that inhibits neuronal activity in response to the exogenous ligand, clozapine-N-oxide[47], AAV2 vectors driven by the CAMKII promoter were used. On postnatal days 40–45, animals were anesthetized (Fluriso), placed in a stereotaxic apparatus and bilateral cannula were used to inject the Gi virus (RAAV2-CAMKIIα-HA-hM4D(Gi)-IRES-mcitrine; 1.4 × 10$^{12}$ vm/ml, 1.0 ul) or the control virus (RAAV2-CAMKIIα-eYFP; 3.8 × 10$^{12}$ vm/ml; 1ul; UNC Vector Core) into each vHipp (A/P + 4.8, M/L ± 4.8, D/V −6.0 mm from bregma).

Five weeks after the virus injections, bilateral guide cannula were implanted directly above the NAc (A/P + 1.4, M/L ± 1.3, DV −6.6 mm from bregma as determined using[70]) or the mPFC (A/P + 3.0, M/L ± 0.6, DV −3.5 mm from bregma as determined using[70]), and secured to the skull using dental cement and four anchor screws. Animals were allowed at least one week to recover before behavioral or electrophysiological experiments were performed. The pathway inhibition strategy is depicted in Fig. 4a, b.

**Whole-cell patch-clamp recordings**. It is well known that ectopic expression of receptors may not necessarily recapitulate the physiological role of that receptor in vivo. To better understand the consequence of a1 and a5 overexpression in vHipp pyramidal neurons, we employed whole-cell patch-clamp electrophysiology. Brains were removed and placed in ice-cold artificial cerebrospinal fluid (aCSF) containing (in mM): 261 sucrose, 2 KCl, 2 MgSO$_4$, 1.25 NaH$_2$PO$_4$, 1 CaCl$_2$, 1 MgCl$_2$, 10 HEPES, 10 glucose, and 0.4 ascorbic acid (pH 7.4, 315 mOsm). Brains were cut into horizontal 300 μm slices with a Vibratome (Leica Microsystems, Wetzlar, Germany). Slices contain the hippocampus were incubated in standard aCSF containing (in mM): 140 NaCl, 2.5 KCl, 2 CaCl$_2$, 1 MgCl$_2$, 10 HEPES, 10 glucose, and 0.4 ascorbic acid (pH 7.4, 295 mOsm) at room temperature for at least 1 h before recording commenced.

Patch-clamp recordings from hippocampal CA1 pyramidal neurons were performed with the aid of IR-DIC optics and a 16-bit EMCCD digital camera (Photometrics, Inc.). Patch electrodes were pulled (Flaming/Brown P-97, Sutter Instrument, Novato, CA, USA) from borosilicate glass capillaries and polished to a tip resistance of 3–5 MΩ. Electrodes were filled with a solution containing (in mM): 120 CsCl, 1 MgCl$_2$, 10 HEPES, 1 EGTA, 4 NaCl, 2 Mg-ATP, 5 QX-314 (pH 7.2, 287 mOsm). Tonic GABA currents and spontaneous IPSC activity were recorded in whole-cell configuration in voltage-clamp mode (V$_{hold}$ = −70 mV) using an Axopatch 200B amplifier and pCLAMP software (v10.3, Axon Instruments, Union City, CA, USA). Signals were filtered at 2 kHz, digitized 10 kHz (Digidata 1440 A, Axon Instruments), and saved on a computer for offline analysis. Recordings were made from neurons identified by the AAV-encoded fluorescent reporter GFP using an appropriate filter set. Recordings were performed with AMPA and NMDA receptors blocked with CNQX (10 μM) and DL-AP5 (50 μM),

respectively. Tonic GABA current was measured as a reduction of holding current following bath application of picrotoxin (100 μM) for ~5 min. Amplitude and frequency of sIPSC activity were quantified using Clampfit software (v10.3) from ~3 min of stable baseline data recorded at least 5 min after achieving whole-cell configuration and prior to picrotoxin exposure.

**Extracellular dopamine recordings**. To measure the activity of dopamine neurons in the ventral tegmental area (VTA), rats were anesthetized with 8% chloral hydrate (400 mg kg$^{-1}$, i.p.), which does not appreciably affect dopamine system function[65]. Rats were then placed in a stereotaxic apparatus. Supplemental anesthesia was administered as necessary and a core body temperature of 37 °C was maintained. Extracellular glass microelectrodes were lowered into the VTA (A/P −5.3, M/L ± 0.6, D/V −6.5 to 9.0 from bregma as determined using[70] using a hydraulic microdrive. Previously established electrophysiological criteria were used to identify spontaneously active dopamine neurons[66] encountered while making 6–9 vertical passes through the VTA. Neuronal activity was filtered (high-frequency cutoff: 30 kHz, low-frequency cutoff: 30 Hz) and recorded using LabChart software (version 7.1; ADInstruments, Chalgrove, Oxfordshire, UK). It should be noted that; while dopamine neurons projecting to different brain regions display distinct electrophysiological signatures[67], the ability to physiologically identify dopamine neurons in vivo has been recently confirmed in a review by Ungless and Grace[68]. The following parameters of dopamine neuron activity were analyzed: (1) population activity, which was defined as the average number of spontaneously activity neurons recorded per electrode track, (2) basal firing rate, and (3) the proportion of action potentials occurring in bursts (bursts defined as the occurrence of two spikes with an interspike interval of 80 ms, and the termination of the burst defined as the occurrence of an interspike interval of 160 ms).

**Extracellular pyramidal cell recordings**. In order to measure putative pyramidal cell activity in the vHipp, we performed in vivo extracellular recordings as described above. Extracellular glass microelectrodes were lowered into the vHipp (A/P −5.0, M/L ± 4.5, D/V −4.0–8.0 from bregma) and putative pyramidal neurons (identified by a firing rate <2 Hz[69]) were recorded. The firing rate was analyzed using Labchart software.

**CNO injections**. On the day of behavioral testing, animals were moved to the behavioral facility and allowed at least 1 h to acclimatize. Approximately 15 min prior to testing, bilateral microinjectors (Plastics One) that extended 1 mm past the end of the indwelling cannula were used to inject 300 uM (0.75 ul, dissolved in saline) CNO into the NAc or mPFC.

**Attentional set-shifting test**. To measure cognitive flexibility, the AST was used. Rats were restricted to 12 g food/day for 7 days prior to testing. Using a cheerio reward, rats were trained to dig in pots defined by cues along two stimulus dimensions: the digging medium filling the pot and an odor (Aura Cacia essential oils) applied to the inner rim of the pot. During testing, the rat was taken through a series of stages, each requiring a different discrimination, with a criterion of six consecutive correct trials required to proceed to the next stage. The first stage was a simple discrimination (SD), with only one stimulus dimension (odor or medium) present. Odor (clove or nutmeg) was the initial relevant dimension (signaling the location of the reward) for half the animals, and medium (raffia or yarn) for the other half. The second stage was a compound discrimination (CD) in which the same discrimination was required and the second irrelevant dimension was introduced. The third stage was a reversal (R1) in which the same odors and media were used, and the same dimension remained relevant, but the negative cue from the previous stage became positive and the positive cue from the previous stage was now negative. Stage 4 was an intradimensional shift (ID), in which all new odors (cinnamon or rosemary) and media (beads or wood balls) were introduced but the same dimension remained relevant. After performing a second reversal (R2), the last stage was an extradimensional set-shift (ED). During ED, all new odors (citronella or thyme) and media (velvet or crepe paper) were presented and the previously irrelevant stimulus became relevant.

**Immunohistochemistry**. Immunohistochemistry was used to confirm virus-mediated gene expression in vHipp neurons. Briefly, after behavioral and physiological experiments were performed, rats were transcardially perfused with saline, followed by 4% paraformaldehyde. Brains were post-fixed and cryoprotected in 10% sucrose in phosphate-buffered saline (PBS). For antigen retrieval, free-floating coronal sections from the vHipp (50 μm) were boiled in 10 mM citric acid, pH 6, for a total of 5 min. Sections were then washed in PBS, blocked (2% normal goat serum and 0.3% Triton X-100), then incubated with chicken anti-GFP antibody (Millipore; 1:1000) at 4 °C overnight. Sections were then incubated with AlexaFluor 488 goat anti-chicken secondary antibody (1:1000). To confirm pyramidal cell expression, a subset of sections were then incubated with mouse anti-CAMKII antibody (Thermo Fisher; 1:200) followed by AlexaFluor 594 goat anti-mouse (1:1000). Sections were mounted on slides and cover-slipped using Prolong gold anti-fade reagent. Sections were imaged using an Olympus IX81 Motorized Inverted microscope. The representative images were acquired using FV10-ASW software and enhanced using ImageJ.

**Statistics**. Animals were randomly assigned to treatment groups. In all figures, data are shown as mean ± SEM and $n$ is indicated in the figure legend. In the patch-clamp experiments, data were analyzed using a one-way ANOVA followed by the Holm–Sidak post-hoc test. For all other experiments, data were analyzed using a two-way ANOVA and Holm–Sidak post-hoc test. All tests were two tailed and significance was determined to at $p < 0.05$.

**Reporting summary**. Further information on research design is available in the Nature Research Reporting Summary linked to this article.

## Data availability
The datasets generated during and/or analyzed during the current study are available from the corresponding author upon reasonable request.

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

## Acknowledgements

This work was supported by the Owens Foundation, R01 MH090067 (D.L.) from the National Institute for Mental Health, P01 HL088052 (G.T.) from the National Heart, Lung and Blood Institute, and by TL1TR002647 (J.D.) from the National Center for Advancing Translational Science. Images were generated in the Core Optical Imaging Facility which is supported by UTHSCSA, NIH-NCI P30 CA54174 (CTRC at UTHSCSA) and NIH-NIA P01AG19316. This content is solely the responsibility of the authors and does not necessarily represent the official views of the National Institutes of Health. We would like to thank Luke Thomas for his technical assistance.

## Author contributions

J.D., D.L. and G.T. participated in research design. J.D., A.B. and J.Y. conducted experiments. J.D., D.L., J.Y. and G.T. performed the data analysis. All authors wrote or contributed to the writing of the manuscript.

## Additional information

**Competing interests:** The authors declare no competing interests.

