## [Peer Review File · Nature Communications]

Reviewers' comments:

Reviewer #1 (Remarks to the Author):

In this manuscript, the authors show data supporting the importance of normalizing hippocampal hyperactivity via modulation of excitability-regulating extrasynaptic GABA A alpha 5 receptors, and importantly that this normalization shows symptom-specific effects depending on the pathway. This is an exciting finding that advances the field. However, some statements are inaccurate or misleading, and should be explained. Specific comments follow:

1. It is unclear why inactivation of the vHipp-mPFC pathway did not attenuate DA neuron population activity. Although this is consistent with the pathway-specific effects of Floresco et al that demonstrated that only pharmacological blockade of the vHipp-NAc pathway normalized DA neuron activity after hippocampal activation, whereas blockade of the vHipp-PFC pathway did not affect population activity, a previous paper from the Lodge group suggested that mPFC activity did impact DA neuron population activity.
2. The social interaction experiment interpretation is confusing. The fact that the MAM rats in the vHipp-PFC study did not show social deficits is not consistent with what has been found in the vHipp-NAc MAM untreated group. The fact that there was no deficit, and that vHipp-mPFC manipulation had no significant effect, suggests that this pathway is not involved in social interactions. To suggest otherwise is misleading. Why didn't the authors test whether the gene therapy alone was sufficient to restore social deficits?
3. While the DA system is involved in mPFC-mediated cognitive flexibility as shown by Robbins et al., this group has also shown that reversal learning is mediated primarily by serotonin tone in the OFC; therefore it is not surprising that vHipp-mPFC manipulations failed to impact reversal learning deficits. In contrast, affecting the vHipp-NAc pathway is likely to impact the OFC-NAc system that Robbins has shown to be involved in reversal learning. Therefore, impacting the OFC-NAc pathway via EITHER normalization of vHipp-NAc activity OR normalization of VTA-NAc activity is a possibility. However, without pharmacological studies it would not be possible to distinguish if DA is indeed involved in the current study.
4. The citation of Lewis et al for schizophrenia loss of interneuron function rather than cell loss is not correct with respect to the target studied. Lewis showed that there is a decrease in PV content per neuron IN THE DLPFC, but did not demonstrate this in the hippocampus. In contrast, both in animal models and in human postmortem studies there is substantial evidence for actual PV neuron loss in the limbic hippocampal region. This would not invalidate the results; increasing the functional impact of the remaining neurons would still be effective, but the way this section is worded is not correct and is misleading.
5. The hypothesis that the anterior hippocampus in humans is functionally equivalent to the ventral hippocampus in rodents is not supported by a citation. This argument has been made effectively in a review by Grace, which could support this statement.
6. The concept that all three symptom domains arise from dysfunction in these areas was actually advanced by O'Donnell (O'Donnell, P. and Grace, A.A. (1998) Dysfunctions in multiple interrelated systems as the neurobiological bases of schizophrenic symptom clusters. *Schizophrenia Bulletin* 24: 267-283) and the involvement of vHipp projections to regions implicated in these symptoms advanced by Gomes (Gomes, F.V. and Grace, A.A. (2016) Prefrontal cortex dysfunction increases susceptibility to schizophrenia-like changes induced by adolescent stress exposure. *Schizophrenia Bulletin* 43: 592-

600.doi: 10.1093/schbul/sbw156). One factor suggested in these papers is that the vHipp-BLA pathway may be involved in negative symptoms, which may account for some discrepant findings.

Reviewer #2 (Remarks to the Author):

This is an interesting and well-written study describing a novel gene therapy for the treatment of schizophrenia symptoms induced by a neurodevelopmental insult. The study builds on the previous observation from the same group that presynaptic restoration of GABA signaling in the ventral hippocampus (through the implantation of stem cells that differentiate into parvalbumin or somatostatin interneurons) can ameliorate preclinical schizophrenia symptoms induced by prenatal exposure to a neurotoxin (methylazoxymethanol). Here the authors instead focus on restoring hippocampus function through a postsynaptic mechanism, by over-expressing the alpha5 subunit of the GABAA receptor. In addition, the authors identify two hippocampal efferents (the nucleus accumbens and medial prefrontal cortex) that are responsible for positive and cognitive schizophrenia-like symptoms. All experiments are well thought out and the methods are sound and appropriate. Even though previous work already confirmed the hypothesis that schizophrenia can be treated by normalizing hippocampus function, the novel use of two different gene therapy strategies that have translational potential for human schizophrenia patients, and the demonstration that distinct disease symptom domains can be differentiated to distinct pathways and treated with pathway-specific manipulations make this an important and exciting study.

1. In the introduction, it is not entirely clear that the studies cited on alpha5 GABA receptor subunit function and the effects of knockdown of this subunit on schizophrenia symptoms were conducted in the hippocampus. It would be helpful to clarify this. It would also be useful if the MAM model, and the effect of prenatal MAM exposure on hippocampal structure and function are briefly introduced. Finally, the concluding statement of the introduction is somewhat confusing given that the authors did not study antipsychotics here.
2. Given that the authors have not looked at the effects alpha5 overexpression on negative schizophrenia-like symptoms, and that the effects of the chemogenetic manipulations did not (significantly) alter social interaction, it would be best to limit the scope of this paper to the positive and cognitive symptom domains of schizophrenia. Although there appears to be a promising trend for an increase in social interaction in the Gi-DREADD vHPC-mPFC rats treated with CNO, no meaningful conclusions can be drawn from this statistically insignificant observation and in light of this, the suggestion that restoring hippocampus function reduces negative symptoms needs to be omitted from the study. In order to incorporate the negative symptom domain into this study, it would be necessary to conduct a social interaction test in alpha5 overexpressing rats and to add more animals to the vHPC-mPFC GFP group to reveal the effects of MAM exposure on reducing social interaction in control animals.
3. Even though CaMKII is a relatively selective promoter for the expression of transgenes into (cortical/allocortical) pyramidal neurons, viral vectors with cell-specific promoters sometimes cause the non-specific expression of transgenes into other cell types. In light of this, it would be important to do a histochemical counterstain for CaMKII on sections from rats infected with the alpha5 lentivirus and look for co-localization. An additional interesting analysis for looking into the specificity of the method (and the effects of pyramidal neuron overexpression of alpha5 on the global physiology of the hippocampus) is to look at effects on the firing rate of putative hippocampal interneurons in the extracellular field recordings. For a better interpretation of the results It would also be helpful to provide an image that covers a larger portion of the (para)hippocampal region to illustrate the amount

of spread of the lentivirus throughout the region.

4. In the methods section, the titer of the eYFP control virus for the chemogenetics studies is very low (3.8×10^4). Was there any expression of the transgene observed in the tissue? Also, what was the volume of CNO solution microinjected into the mPFC/NAc and what was the vehicle that was used to dissolve the drug (e.g. saline or aCSF). For the immunohistochemical staining procedure, how long were sections boiled in the retrieval buffer? Finally, were rats housed under a normal day/night cycle?

5. Figure 4 does not show that "Chemogenetics can be used to inactivate the different afferents from the vHPC". Perhaps it would be better to incorporate the cartoons and representative images into Figs. 5 and 6. In addition, it would be useful to include a low magnification micrograph of the (para)hippocampal region to illustrate which subregions were infected with the DREADDs. Was the expression restricted to the CA1-CA3 or did it also include the DG, subiculum, and other parahippocampal structures? The spread of infection might have implications for the interpretation of the data, given that both the CA1 and subiculum project to the NAc and PFC but likely convey different information. Finally, which subregions of the mPFC and NAc were targeted by the cannulas? Please include representative micrographs of the terminal fields in the mPFC and NAc, and include a figure containing the location of cannulas or microinjection needle tips in the Core/Shell or ACC/PL/IL subareas of the NAc and mPFC for all animals.

6. Although the manuscript is very well written, it would be helpful if the authors could do an additional screen for errors in spelling and sentence structure. For instance in the introduction: "Using chemogenetics, we identified >the< discrete afferent pathways from the vHipp mediate the behavioral and physiological deficits that ...". and in the results: "... MAM-treated rats required significantly >higher trials< to reach criterion than ...". In addition, please double check the references, for instance the human gene therapy study for PD mentioned in the discussion was done with striatal neurons, not in nigrostriatal neurons.

7. The discussion section is a bit lengthy and it would be useful to reduce the size of specific paragraphs and remove parts that are not directly relevant. For instance, the discussion on the use of intracranial CNO and off-target effects of CNO are not directly relevant for the interpretation of the data in this paper. Perhaps the rationale for using microinjected CNO can instead be briefly mentioned in the results section.

8. What is the authors' explanation for the observation that alpha5 overexpression did not normalize reversal learning deficits, but chemogenetically inhibiting the vHPC-NAc did. Could the alpha5 vector somehow preferentially alter the function of vHPC-mPFC projecting neurons? Could this be attributed to a difference in the localization of the lentivirus and the AAV in the HPC? Please include a brief discussion of this discordance in the study.

Reviewer #3 (Remarks to the Author):

This is an interesting manuscript by Donegan et al. where the authors rescue dopamine neuron hyperactivity and cognitive behaviors in the MAM model by GABA-A alpha 5 receptors overexpression in the ventral hippocampus or by chemogenetic inhibition of vHipp-mPFC or vHipp-NAc projections.

The results may help to understand circuit alterations observed in patients with schizophrenia. What surprises me is that the GABA-A alpha 5 overexpression and the DREADD pathway inhibitions are only having an effect on in vivo physiology and behavior in the MAM model but not in wild-type controls

(with the exception of burst firing for the vHipp-mPFC inhibition). For the alpha5 overexpression this is despite strongly enhanced tonic GABA currents measured in brain slices of wild-type mice. It would be important to understand at the mechanistic level why the MAM model is more sensitive to the inhibition experiments. By now there have been several publications exploring the behavioral and DA neuron phenotypes in the MAM model. The increase in mechanistic insight by the presented data here is relatively limited.

Major points:

Figure 1: I am surprised that there is no effect of alpha5 on sIPSCs as in the CA1 area perisynaptic $\alpha 5$ -GABAARs have been shown to contribute to slow phasic currents (e.g. Prenosil et al. 2006). Also, it is unclear to me why sIPSC amplitude is not increased after GABA-A receptor overexpression. This would need to be explained.

Figure 1: How do the authors explain that alpha5 overexpression has suppress hippocampal in vivo extracellular activity only in the MAM model but not in wild-type mice? This is despite the large effect that alpha5 overexpression has on tonic GABA currents in the patch clamp experiments. Mechanistic insights for this specificity would be needed.

Figure 2: It is now relatively well established that electrophysiological criteria are limited in distinguishing DA from non-DA neurons in the VTA (e.g. Stephan Lammel's work). The limitations of the DA neuron recordings presented here need to be discussed. Moreover, all in vivo recordings are done in anesthetized animals and therefore linking the physiological abnormalities to the behavior is difficult.

Relevance to schizophrenia: The authors study NAC to VTA circuitry in the regulation of VTA DA neuron activity with the argument that this is important for understanding positive symptoms. However, PET imaging studies by Oliver Howes and Anissa Abi-Dargham indicate that DA hyperactivity linked to positive symptoms is mostly seen in the associative striatum and not the limbic NAc. Thus, the relevance of VTA-NAc DA hyperactivity for schizophrenia is not straight forward and would need to be discussed.

The authors revert DA-hyperactivity and cognitive deficits in the animal model using the same rescue manipulation; decreasing hippocampal hyperactivity. However, in schizophrenia patients positive and cognitive symptoms do not correlate well in patients suggesting independent underlying mechanisms. Is the idea that distinct hippocampal projections are differently affected in different patients?

Minor points:

Figure 1A: Is this an IHC against alpha5 or EGFP from the alpha5 ires EGFP construct?

Figure 1B: How did the authors identify alpha5 overexpressing neurons? The method section states that the authors recorded from mCherry positive neurons but the alpha5 lentivirus expresses EGFP.

Calling the rat rescue experiments "gene therapy" experiments is a little bit of an oversell. Gene therapy is usually done in humans and not rodents.

Figure 3A: the bar indicating significance is misplaced.

There is a pronounced effect of vHipp-mPFC inhibition on DA neuron burst firing in the MAM model and in controls. However, decreased DA burst firing seems not to affect attentional set shifting. This would benefit from a discussion.

Reviewer #1

Specific Comments

1. It is unclear why inactivation of the vHipp-mPFC pathway did not attenuate DA neuron population activity. Although this is consistent with the pathway-specific effects of Floresco et al that demonstrated that only pharmacological blockade of the vHipp-NAc pathway normalized DA neuron activity after hippocampal activation, whereas blockade of the vHipp-PFC pathway did not affect population activity, a previous paper from the Lodge group suggested that mPFC activity did impact DA neuron population activity.

Response: The reviewer is correct that we have shown previously (Lodge 2011) that synchronous cortical burst firing of the mPFC can increase the activity of dopamine neurons in the VTA. However, tonic changes in mPFC activity are more nuanced and more likely reflect that seen in the current experiments where neither saline- nor MAM-treated rats are experiencing high levels of coordinated mPFC hyperexcitability, especially under anesthesia. This is likely why we did not observe an effect of vHipp-mPFC inhibition on dopamine population activity in the VTA. We have added a few sentences to the discussion section to clarify this point.

2. The social interaction experiment interpretation is confusing. The fact that the MAM rats in the vHipp-PFC study did not show social deficits is not consistent with what has been found in the vHipp-NAc MAM untreated group. The fact that there was no deficit, and that vHipp-mPFC manipulation had no significant effect, suggests that this pathway is not involved in social interactions. To suggest otherwise is misleading. Why didn't the authors test whether the gene therapy alone was sufficient to restore social deficits?

Response: We agree that the social interaction data was confusing. We have observed previously (Donegan 2018) that manipulations of the prefrontal cortex can reduce the magnitude of the MAM effect. Further, we did test the effect of gene therapy on social interaction but the results were inconclusive. At the suggestion of Reviewer #2, we have decided to remove the social interaction data from the manuscript and focus on positive symptoms and cognitive deficits.

3. While the DA system is involved in mPFC-mediated cognitive flexibility as shown by Robbins et al., this group has also shown that reversal learning is mediated primarily by serotonin tone in the OFC; therefore it is not surprising that vHipp-mPFC manipulations failed to impact reversal learning deficits. In contrast, affecting the vHipp-NAc pathway is likely to impact the OFC-NAc system that Robbins has shown to be involved in reversal learning. Therefore, impacting the OFC-NAc pathway via EITHER normalization of vHipp-NAc activity OR normalization of VTA-NAc activity is a possibility. However, without pharmacological studies it would not be possible to distinguish if DA is indeed involved in the current study.

Response: We agree that other brain regions, namely the OFC, have been strongly implicated in reversal learning and that our effects may result from changes in the OFC-NAc connection, rather than via an increase in dopamine signaling. Therefore, we have expanded the Discussion section to outline this possibility.

4. The citation of Lewis et al for schizophrenia loss of interneuron function rather than cell loss is not correct with respect to the target studied. Lewis showed that there is a decrease in PV content per neuron IN THE DLPFC, but did not demonstrate this in the hippocampus. In contrast, both in animal models and in human postmortem studies there is substantial evidence for actual PV neuron loss in the limbic hippocampal region. This would not invalidate the results; increasing

the functional impact of the remaining neurons would still be effective, but the way this section is worded is not correct and is misleading.

Response: We apologize for this oversight. In the introduction, instead of using the Lewis reference that focuses on the PFC, we have cited manuscripts by Zhange and Reynolds as well as Heckers and Konradi to support our claim that GABAergic dysfunction has been observed in the hippocampus of schizophrenia patients. Further, we have re-worded this section to more clearly state that an actual loss of cells, rather than just a loss of PV expression, has been observed in the hippocampus.

5. The hypothesis that the anterior hippocampus in humans is functionally equivalent to the ventral hippocampus in rodents is not supported by a citation. This argument has been made effectively in a review by Grace, which could support this statement.

Response: We have now included multiple citations to support this statement, including the Grace 2012 review.

6. The concept that all three symptom domains arise from dysfunction in these areas was actually advanced by O'Donnell (O'Donnell, P. and Grace, A.A. (1998) Dysfunctions in multiple interrelated systems as the neurobiological bases of schizophrenic symptom clusters. *Schizophrenia Bulletin* 24: 267-283) and the involvement of vHipp projections to regions implicated in these symptoms advanced by Gomes (Gomes, F.V. and Grace, A.A. (2016) Prefrontal cortex dysfunction increases susceptibility to schizophrenia-like changes induced by adolescent stress exposure. *Schizophrenia Bulletin* 43: 592-600.doi: 10.1093/schbul/sbw156). One factor suggested in these papers is that the vHipp-BLA pathway may be involved in negative symptoms, which may account for some discrepant findings.

Response: We apologize for the oversight and have now expanded the Discussion to credit to the O'Donnell and Grace (1998) manuscript for first proposing the idea that the three primary symptom domains result from dysfunction in the brain regions examined. In addition, we have included the Gomes, *et al* (2016) reference to support the idea that under some conditions, mPFC manipulations can influence dopamine signaling (in line with the first comment). Finally, we included a brief discussion of the role of the BLA-vHipp pathway in schizophrenia-like behaviors.

Reviewer #2

1. In the introduction, it is not entirely clear that the studies cited on alpha5 GABA receptor subunit function and the effects of knockdown of this subunit on schizophrenia symptoms were conducted in the hippocampus. It would be helpful to clarify this. It would also be useful if the MAM model, and the effect of prenatal MAM exposure on hippocampal structure and function are briefly introduced. Finally, the concluding statement of the introduction is somewhat confusing given that the authors did not study antipsychotics here.

Response: The introduction section has been modified to clarify that the alpha 5 agonist was systemic while the knock-down was targeted to the hippocampus. We have also added a sentence to introduce the MAM model and its effects on the hippocampus. Finally, we have modified the final sentence to say, "Together, these experiments explore a novel approach for treating schizophrenia and provide insight into the neuronal pathways associated with discrete dimensions of schizophrenia, so that therapeutics can be developed with improved efficacy at treating multiple symptom domains."

2. Given that the authors have not looked at the effects alpha5 overexpression on negative schizophrenia-like symptoms, and that the effects of the chemogenetic manipulations did not (significantly) alter social interaction, it would be best to limit the scope of this paper to the positive and cognitive symptom domains of schizophrenia. Although there appears to be a promising trend for an increase in social interaction in the Gi-DREADD vHPC-mPFC rats treated with CNO, no meaningful conclusions can be drawn from this statistically insignificant observation and in light of this, the suggestion that restoring hippocampus function reduces negative symptoms needs to be omitted from the study. In order to incorporate the negative symptom domain into this study, it would be necessary to conduct a social interaction test in alpha5 overexpressing rats and to add more animals to the vHPC-mPFC GFP group to reveal the effects of MAM exposure on reducing social interaction in control animals.

Response: We agree with this comment and that of Reviewer 1, who thought the social interaction data was confusing. Therefore, we have removed the social interaction data from the manuscript and now focus on the behavioral correlates of positive and cognitive symptoms.

3. Even though CaMKII is a relatively selective promoter for the expression of transgenes into (cortical/allocortical) pyramidal neurons, viral vectors with cell-specific promoters sometimes cause the non-specific expression of transgenes into other cell types. In light of this, it would be important to do a histochemical counterstain for CaMKII on sections from rats infected with the alpha5 lentivirus and look for co-localization. An additional interesting analysis for looking into the specificity of the method (and the effects of pyramidal neuron overexpression of alpha5 on the global physiology of the hippocampus) is to look at effects on the firing rate of putative hippocampal interneurons in the extracellular field recordings. For a better interpretation of the results It would also be helpful to provide an image that covers a larger portion of the (para)hippocampal region to illustrate the amount of spread of the lentivirus throughout the region.

Response: We understand the reviewer's concern about the specificity of gene expression and have now completed additional immunohistochemistry experiments to confirm that GFP expression was indeed limited to CAMKII-positive cells. Representative images are included in Figure 1. While the reviewer is correct that non-specific expression of the alpha 5 receptor in interneurons may influence the global physiology of the hippocampus, we have demonstrated both an increase in tonic GABA current and decreased firing rates in pyramidal cells, suggesting increased inhibition in these cells. Further, we have also measured the spread of the lentivirus infection, which has been included in the results section of the text.

4. In the methods section, the titer of the eYFP control virus for the chemogenetics studies is very low (3.8×10^4). Was there any expression of the transgene observed in the tissue? Also, what was the volume of CNO solution microinjected into the mPFC/NAc and what was the vehicle that was used to dissolve the drug (e.g. saline or aCSF). For the immunohistochemical staining procedure, how long were sections boiled in the retrieval buffer? Finally, were rats housed under a normal day/night cycle?

Response: We apologize for the error but the titer for the control virus was 3.8×10^{12} and YFP expression was observed for both the control and alpha 5 viruses. For the DREADD experiments, CNO was dissolved in saline and 0.75ul was injected into each hemisphere. For the immunohistochemistry, sections were boiled for 5 minutes total. Finally, rats were housed on a 12h/12h light/dark cycle. We have modified the methods section to include this information.

5. Figure 4 does not show that "Chemogenetics can be used to inactivate the different afferents from the vHPC". Perhaps it would be better to incorporate the cartoons and representative images into Figs. 5 and 6. In addition, it would be useful to include a low magnification

micrograph of the (para)hippocampal region to illustrate which subregions were infected with the DREADDs. Was the expression restricted to the CA1-CA3 or did it also include the DG, subiculum, and other parahippocampal structures? The spread of infection might have implications for the interpretation of the data, given that both the CA1 and subiculum project to the NAc and PFC but likely convey different information. Finally, which subregions of the mPFC and NAc were targeted by the cannulas? Please include representative micrographs of the terminal fields in the mPFC and NAc, and include a figure containing the location of cannulas or microinjection needle tips in the Core/Shell or ACC/PL/IL subareas of the NAc and mPFC for all animals.

Response: We have now incorporated the images from Figure 4 into Figure 5. We have also added cartoons and representative micrographs to show the localization of the CNO injection. In the current experiments we did not target specific areas of the NAc, mPFC, or vHipp but rather injected in a location that would target all subregions. In future experiments we will parse out more specific pathways that are involved in schizophrenia-like behaviors.

6. Although the manuscript is very well written, it would be helpful if the authors could do an additional screen for errors in spelling and sentence structure. For instance in the introduction: "Using chemogenetics, we identified >the< discrete afferent pathways from the vHipp mediate the behavioral and physiological deficits that ..." and in the results: "... MAM-treated rats required significantly >higher trials< to reach criterion than ...". In addition, please double check the references, for instance the human gene therapy study for PD mentioned in the discussion was done with striatal neurons, not in nigrostriatal neurons.

Response: We apologize for the oversights. We have corrected the specific errors listed above and performed an additional screen of the manuscript for other errors or typos.

7. The discussion section is a bit lengthy and it would be useful to reduce the size of specific paragraphs and remove parts that are not directly relevant. For instance, the discussion on the use of intracranial CNO and off-target effects of CNO are not directly relevant for the interpretation of the data in this paper. Perhaps the rationale for using microinjected CNO can instead be briefly mentioned in the results section.

Response: We thank the reviewer for this feedback. We have made the Discussion section more concise and limited the discussion to points that are directly relevant to the current manuscript.

8. What is the authors' explanation for the observation that alpha5 overexpression did not normalize reversal learning deficits, but chemogenetically inhibiting the vHPC-NAc did. Could the alpha5 vector somehow preferentially alter the function of vHPC-mPFC projecting neurons? Could this be attributed to a difference in the localization of the lentivirus and the AAV in the HPC? Please include a brief discussion of this discordance in the study.

Response: We were also surprised to find that alpha 5 over-expression did not normalize reversal learning deficits while chemogenetic inhibition of the vHipp-NAc pathway did. There is strong evidence suggesting that reversal learning depends on dopamine signaling in the striatum. However, reversal learning has also been associated with serotonin signaling in the orbital frontal cortex, a region of the brain that strongly innervates dorsal and ventral striatum. Therefore, it is possible that the alpha 5 over-expression strategy and the vHipp-NAc inhibition strategy differentially affected pathways downstream of the NAc. This possibility has been added to the Discussion section.

Reviewer #3

1. Figure 1: I am surprised that there is no effect of alpha5 on sIPSCs as in the CA1 area perisynaptic $\alpha 5$ -GABAARs have been shown to contribute to slow phasic currents (e.g. Prenosil et al. 2006). Also, it is unclear to me why are sIPSC amplitude is not increased after GABA-A receptor overexpression. This would need to be explained.

Response: The reviewer is correct that previous studies have shown (in knock-out animals) that GABA_A $\alpha 5$ receptors can contribute to slow phasic currents in the CA1 region of the hippocampus. The reason we performed *in vitro* experiments was to determine the effect of viral-mediated over-expression that may or may not replicate those seen under physiological conditions. Here we clearly demonstrate that over-expression of GABA_A $\alpha 5$ receptor produces effects on tonic inhibition and ventral hippocampal pyramidal cell activity. This is now clarified in the paper.

2. Figure 1: How do the authors explain that alpha5 overexpression has suppress hippocampal *in vivo* extracellular activity only in the MAM model but not in wild-type mice? This is despite the large effect that alpha5 overexpression has on tonic GABA currents in the patch clamp experiments. Mechanistic insights for this specificity would be needed.

Response: It is possible that in the *in vivo* extracellular electrophysiology experiments, we are observing a floor effect in the pyramidal cell firing rates, which would explain why alpha 5 over-expression does not affect firing rates in saline-treated animals. Also, it is important to note that in the patch clamp experiments, we recorded only transfected cells, which were identified by GFP fluorescence. However, we did not have the capability to specifically target the transfected cells *in vivo*, so all putative pyramidal cells were recorded. This could explain why alpha 5 over-expression increases tonic GABA currents in saline-treated animals *in vitro* but has no effect on the firing rate of pyramidal cells *in vivo*.

3. Figure 2: It is now relatively well established that electrophysiological criteria are limited in distinguishing DA from non-DA neurons in the VTA (e.g. Stephan Lammel's work). The limitations of the DA neuron recordings presented here need to be discussed. Moreover, all *in vivo* recordings are done in anesthetized animals and therefore linking the physiological abnormalities to the behavior is difficult.

Response: We agree with the reviewer in that the 2008 Neuron article by Lammel, et al provide elegant data that indicate dopamine neurons display distinct electrophysiological signals based on their projection target. However, these data demonstrate not only that dopamine projections to the NAc shell and striatum display long duration action potentials, but that dopamine neurons projecting to the mPFC, BLA, and NAc core and medial shell are significantly longer. The ability to physiologically identify dopamine neurons *in vivo* has been recently addressed in a review by Ungless and Grace, aptly titled "Are you or aren't you? Challenges associated with physiologically identifying dopamine neurons." This important point is now discussed in the revised manuscript. Further, although we cannot conclusively link the change in dopamine activity to behavior without measuring conscious recordings, chloral hydrate was specifically selected as an anesthetic as it does not dramatically alter dopamine neuron activity. A sentence addressing this has been added to the methods section.

4. Relevance to schizophrenia: The authors study NAC to VTA circuitry in the regulation of VTA DA neuron activity with the argument that this is important for understanding positive symptoms. However, PET imaging studies by Oliver Howes and Anissa Abi-Dargham indicate that DA

hyperactivity linked to positive symptoms is mostly seen in the associative striatum and not the limbic NAc. Thus, the relevance of VTA-NAc DA hyperactivity for schizophrenia is not straight forward and would need to be discussed.

Response: We agree with the reviewer that there are likely differences between schizophrenia patients, in which recent studies have demonstrated the largest dopamine aberrations in the dorsal striatum, and animal models of positive symptoms, which have observed more robust dopamine alterations in the ventral striatum. Therefore, we have added a caveat to the discussion section.

5. The authors revert DA-hyperactivity and cognitive deficits in the animal model using the same rescue manipulation; decreasing hippocampal hyperactivity. However, in schizophrenia patients positive and cognitive symptoms do not correlate well in patients suggesting independent underlying mechanisms. Is the idea that distinct hippocampal projections are differently affected in different patients?

Response: The reviewer is correct that schizophrenia is a very heterogeneous disorder and that different patients may have distinct pathophysiology. We are not proposing that this vHipp pathway is responsible for all aspects of schizophrenia in all patients; however, it is a known site of pathology that may contribute to distinct symptoms domains. This is now clarified in the Discussion.

6. Figure 1A: Is this an IHC against alpha5 or EGFP from the alpha5 ires EGFP construct?

Response: In the IHC experiments, we used an antibody against GFP to identify gene expression in both control and alpha 5 over-expression animals. This has been clarified in the figure legend.

7. Figure 1B: How did the authors identify alpha5 overexpressing neurons? The method section states that the authors recorded from mCherry positive neurons but the alpha5 lentivirus expresses EGFP.

Response: We apologize for this error. The reviewer is correct that the virus expresses GFP, which is the fluorescent reporter that we used to identify alpha 5 over-expressing neurons.

8. Calling the rat rescue experiments “gene therapy” experiments is a little bit of an oversell. Gene therapy is usually done in humans and not rodents.

Response: Although we agree that “gene therapy” usually refers to a human treatment, in the current experiments, we specifically focused on a gene and viral vector that could be translated into this treatment strategy. However, we understand the concern of the reviewer and have tempered the language throughout.

9. Figure 3A: the bar indicating significance is misplaced.

Response: The bar was placed to demonstrate a significant main effect of prenatal treatment. The figure legend has been updated to indicate that the significance was in the main effect.

10. There is a pronounced effect of vHipp-mPFC inhibition on DA neuron burst firing in the MAM model and in controls. However, decreased DA burst firing seems not to affect attentional set shifting. This would benefit from a discussion.

Response: This result was somewhat surprising to us as well. We have now added a sentence to the Discussion section suggesting that it is possible that the change in dopamine signaling caused by vHipp-mPFC inhibition did not change dopamine signaling in the caudate, the specific striatal sub-region implicated in reversal learning.

REVIEWERS' COMMENTS:

Reviewer #1 (Remarks to the Author):

The authors did a comprehensive job of responding to all of my critiques. I have no further comments.

Reviewer #2 (Remarks to the Author):

The authors have addressed all my points satisfactorily, and the manuscript has been much improved. I do not have any additional concerns, and recommend acceptance without further revision.

Reviewer #3 (Remarks to the Author):

The authors addressed all of my concerns.